# Aqueous Chemical Synthesis of Nanosized ZnGa$_2$O$_4$ Using Mild Reaction Conditions: Effect of pH on the Structural, Morphological, Textural, Electronic, and Photocatalytic Properties

Dalia Téllez-Flores [1], Manuel Sánchez-Cantú [1,*], María de Lourdes Ruiz-Peralta [1], Esteban López-Salinas [2], Armando Pérez-Centeno [3], Francisco Tzompantzi [4] and Alejandro Escobedo-Morales [1,*]

[1] Facultad de Ingeniería Química, Benemérita Universidad Autónoma de Puebla, Avenida San Claudio y 18 Sur, Puebla 72570, Mexico; dalia.tellez@alumno.buap.mx (D.T.-F.); lourdes.ruiz@correo.buap.mx (M.d.L.R.-P.)
[2] Instituto Mexicano del Petróleo, Eje Central Lázaro Cárdenas 152, México City 07730, Mexico; esalinas@www.imp.mx
[3] Departamento de Física, Centro Universitario de Ciencias Exactas e Ingenierías, Universidad de Guadalajara, Boulevard Marcelino García Barragán 1421, Guadalajara 44430, Mexico; armando.centeno@cucei.udg.mx
[4] Departamento de Química, Área de Catálisis, Universidad Autónoma Metropolitana Iztapalapa, Av. San Rafael Atlixco 189, Iztapalapa, México City 09340, Mexico; fjtz@xanum.uam.mx
[*] Correspondence: manuel.sanchez@correo.buap.mx (M.S.-C.); alejandro.escobedo@correo.buap.mx (A.E.-M.); Tel.: +52-(222)-2295500 (ext. 7265) (M.S.-C. & A.E.-M.)

**Abstract:** The effect of pH on the structural, textural, morphological, and electronic properties of ZnGa$_2$O$_4$ nanoparticles obtained by coprecipitation using mild reaction conditions (25 °C; 30 min) was studied. The pH ranges in which coprecipitation reactions occurred and the chemical species associated with the reaction mechanism were identified. It was determined that the samples synthesized at pH values between 6 and 10 consisted of Zn-Ga oxide blends, with spinel ZnGa$_2$O$_4$ being the majority phase. Conversely, the material prepared at pH 12 was constituted by Zn-Ga layered double hydroxide phase along with wurtzite ZnO traces. The synthesis pH determined the reaction product yield, which decreased from 51 to 21% when the reaction medium turned from softly acidic (pH 6) to strongly alkaline conditions (pH 12). The bandgap energies of the synthesized materials were estimated to be in the range of 4.71–4.90 eV. A coprecipitation-dissolution-crystallization mechanism was proposed from the precipitation curve, with specific mononuclear and polynuclear species being involved in the formation of the different precipitates. Phenol was employed as a probe molecule to evaluate the photocatalytic performance of the synthesized samples. Among the samples, the one prepared at pH 6 showed the largest photodegradation efficiency (~98%), which was superior to commercial TiO$_2$-Degussa P25 (~88%) under the same process conditions, which can be attributed to both its high specific surface area (140 m$^2$ g$^{-1}$) and the formation of a Zn$_{2x}$Ga$_{2-2x}$O$_{3+x}$/ZnGa$_2$O$_4$ heterojunction.

**Keywords:** spinel; zinc gallate; layered double hydroxide; photocatalysis

## 1. Introduction

Zinc gallate (ZnGa$_2$O$_4$) is a ternary oxide that crystallizes in a spinel structure (AB$_2$O$_4$). It is a *p*-type conducting compound with a wide bandgap (~5 eV), and belongs to the transparent semiconductors group [1]. Due to their atomic arrangement, a diversity of cations can be incorporated into the structure of spinel-like materials to obtain compounds with controlled properties. Owing to its chemical and thermal stability, zinc gallate is considered a promising material for optoelectronic and power devices for use in panel displays, vacuum fluorescent displays, day-blind detectors, fluorescent materials, deep-ultraviolet photodetectors, phototransistors, gas sensors, thin film-transistors [2], and in

photocatalytic processes. In this regard, it has been evaluated in photocatalytic hydrogen production, benzene decomposition, $CO_2$ reduction, and water splitting. Additionally, it has been tested in the photodegradation of several textile dyes, including rhodamine B, methyl orange, and methylene blue [3].

Among the conventional synthesis methods for obtaining $ZnGa_2O_4$ are solid-state reaction [4], sol-gel [5], hydrothermal [6], combustion [7], and metal organic chemical vapor deposition (MOCVD) [8]. However, these approaches frequently involve high reaction temperatures (~1200 °C), long synthesis times (24 h) [9], and/or expensive equipment. For instance, Hirano et al. [10] reported the synthesis of $ZnGa_2O_4$ through calcination of powders obtained by coprecipitation of zinc sulphate and gallium sulphate. These authors demonstrated that the precursor ratio, nature of the precipitating agent, and aging temperature influence the microstructure of the synthesized materials. However, additional studies concerning the effects of other chemical parameters that determine not just the resulting phases but the reaction yield need to be conducted as well. In this matter, the pH of the reaction medium has paramount importance for gallium-based compounds, as the incorporation of the metalloid element is desired as large as possible. For instance, although $Ga_2O_3$ has shown interesting results as photocatalyst by itself [11,12], doped gallium oxide samples have demonstrated higher catalytic activity than their pristine counterparts [13]. In this regard, Sakata et al. obtained $Ga_2O_3$ impurified with different metal elements and observed that doping with Zn improves the photocatalytic activity of gallium oxide. The authors assigned this behavior to formation of $ZnGa_2O_4$ compound, which has been used successfully for dye degradation [3], though not for water splitting [14].

Here, the synthesis of $ZnGa_2O_4$ powders by aqueous soft coprecipitation of zinc and gallium nitrates under acid or basic conditions applying short aging times is presented. This parameter is crucial to obtaining superior textural properties by reducing the aggregation rate and growth of the $ZnGa_2O_4$ particles. The importance of the effect of pH [15] on the structure, morphology, optical, electronic, and textural properties of the obtained materials was studied. Finally, the synthesized materials were evaluated as photocatalysts, using phenol as probe molecule, and their performance was correlated with the microstructure features resulting from different synthesis conditions.

## 2. Materials and Methods

### 2.1. Materials and Reagents

Zinc nitrate hexahydrate ($Zn(NO_3)_2 \cdot 6H_2O$, 99.0%; Meyer, México City, Mexico), gallium nitrate nonahydrate ($Ga(NO_3)_3 \cdot 9H_2O$, 98.9%; Johnson Mattey, Royston, UK), potassium hydroxide (KOH, 85.0%; Fermont, Monterrey, Mexico), and phenol ($C_6H_5OH$, 99.5%; Mallinckrodt Pharmaceuticals, St. Louis, MO, USA) were purchased and used as received without further purification. Deionized and decarbonated water was employed in all the experiments.

### 2.2. Synthesis of Materials

The materials were synthesized through a soft aqueous coprecipitation method at room temperature [16] following the procedure shown in Figure 1. Initially, zinc nitrate and gallium nitrate were dissolved in decarbonated, deionized, and distilled water to obtain a 0.5 Zn/Ga molar ratio precursor solution. A 0.5 M KOH aqueous solution was prepared as precipitant, then specific volumes of the previous solutions were mixed until the pH of the reaction medium reached values of 6, 8, 10, and 12. Afterwards, the mixture was aged at room temperature for 30 min under magnetic stirring. Finally, the obtained precipitate was washed with hot water (80 °C), centrifuged at 5000 rpm for 15 min, and dried in air at 120 °C for 2 h. The samples were labelled as S6, S8, S10, and S12, where the numbers represent the pH value employed during synthesis.

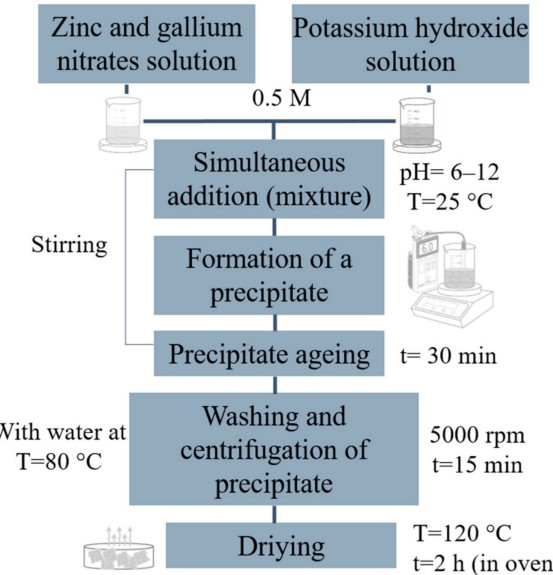

**Figure 1.** Flowchart of the synthesis method.

### 2.3. Titration Study

A titration study was conducted to identify the precipitation and coprecipitation zones of the metallic cations, following the protocol described by Ross and Kodama [17]. Briefly, 40 mL of a 0.5 M KOH solution was added dropwise to 10 mL of the Zn-Ga precursor solution under magnetic stirring, then the titrant volume versus the pH of the mixture was recorded.

### 2.4. Photocatalysis Experiments

The photocatalytic activity of selected samples was assessed via phenol degradation. All the experiments were performed in a Pyrex batch photoreactor equipped with a cooling system to keep the reaction medium at room temperature. A UVP Pen-Ray power supply model 11SC-1 (254 nm, 4.4 mW cm$^{-2}$; analytikjena Co., Upland, CA, USA) was used as light source. In a standard experiment, a suspension comprising the phenol solution and the catalyst was stirred in dark during 30 min until adsorption–desorption equilibrium was achieved. Afterwards, 200 mg of the photocatalyst was dispersed in 200 mL of aqueous phenol solution ($4 \times 10^{-2}$ g L$^{-1}$) and the UV light was turned on. During the entire experiment (360 min), a continuous flux of air was bubbled into the solution. Aliquots were periodically withdrawn from the reaction medium every 30 min; the catalyst was separated from the liquid phase using a syringe-filter PTFE (polytetrafluoroethylene) membrane. The concentration of phenol was determined by spectrophotometry using a Cary-100 UV–Vis spectrophotometer (Agilent Technologies, Santa Clara, CA, USA) and recording the optical absorbance of phenol at 270 nm wavelength. The photocatalysis efficiency was calculated as follows:

$$\eta = \frac{C_0 - C}{C_0} \times 100\% \tag{1}$$

where $C_0$ is the initial phenol concentration and $C$ is the concentration of phenol at a given time [18]. The obtained samples were compared with TiO$_2$-Degussa P25 as a de facto standard titania photocatalyst. Photolysis of phenol molecules under UV light was measured under analogous experimental conditions.

### 2.5. Total Organic Carbon

Total organic carbon (TOC) was measured using a TOC-V-CSH/CSN analyzer (Shimadzu Co., Kyoto, Japan) to determine mineralization of organic carbon in the aqueous media.

### 2.6. Materials Characterization

The synthesized samples were analyzed by powder X-Ray diffraction (XRD) using an Advance D-8 diffractometer (Bruker Co., Berlin, Germany) equipped with a Cu-K$\alpha$ radiation source. The diffracted X-Rays were scanned in the 2θ range of 5°–70° with a step size and counting time of 0.04° and 0.5 s per step, respectively. The morphology of the samples was examined by field emission scanning electron microscopy (FE-SEM) using a Mira 3 LMU electron microscope (Tescan Ltd., Brno, Czech Republic). For this purpose, the samples were mounted on double-sided adhesive carbon tape and the micrographs were acquired by setting the accelerating voltage at 20 kV. The optical properties of the materials were studied by diffuse reflectance spectroscopy (DRS) by means of a Cary-5000 UV–Vis spectrophotometer (Agilent Technologies, Santa Clara, CA, USA) equipped with a DRA-CA-30I integration sphere. Textural analysis was carried out using an Autosorb-3B automatic surface area and pore size analyzer (Quantachrome Instruments, Boynton Beach, FL, USA).

### 3. Results and Discussion

#### 3.1. Microstructure

Figure 2 shows the XRD patterns of the samples synthesized at different pH values. A comparison among them reveals that the pH of the reaction medium influences the resulting crystalline phase. Specifically, all the X-ray reflections of samples S6, S8, and S10 match that of the $ZnGa_2O_4$ spinel structure (JCPDS 71-0843). However, a polyphasic sample containing zinc–gallium layered double hydroxide (Zn-Ga LDH; JCPDS 38-0486) mixed with zinc oxide (ZnO; JCPDS 89-1397) and zinc gallate ($ZnGa_2O_4$; JCPDS 71-0843) as minority phases occurs under stronger alkaline conditions (pH 12). Interestingly, formation of Zn-Ga LDH by coprecipitation has been reported at lower pHs as well [19]. However, its formation depends on the molar ratio of $Zn^{2+}$ and $Ga^{3+}$. For instance, Hu et al. [20] synthesized a series of Zn-Ga LDH samples using different Zn:Ga molar ratios under soft alkaline conditions (pH 8). They noticed that if a Zn:Ga molar ratio of one or two is used, a small amount of $ZnGa_2O_4$ results. These authors attributed the formation of the spinel phase to the zinc and gallium species present at this pH. Additionally, they pointed out that as the Zn:Ga ratio increases, larger amounts of Zn-Ga LDH can be achieved. These results suggest that the nature and concentration of the chemical species can be controlled by adjusting the pH of the reaction medium, meaning that the resulting phases can be as well.

The calculated lattice parameters of the phases identified by XRD are summarized in Table 1. The value of the lattice parameter of the S6 sample corresponds with the standard value of $ZnGa_2O_4$ (JCPDS 71-0843). However, the X-Ray peaks shift slightly towards lower diffraction angles as the pH increases, indicating a larger unit cell. In the case of $Ga^{3+}$ cations, the octahedral coordination is larger than the tetrahedral coordination of $Zn^{2+}$ cations (76 vs. 74 pm, respectively). For this reason, $Ga^{3+}$ cations cannot easily be incorporated as $Zn^{2+}$ into the spinel structure. Then, the increase of the lattice parameter might be attributed to interstitial Zn defects [21]. In the case of S12 material, the *a* lattice parameter again agrees with the standard value. This result is expected, as molar ratios of divalent versus trivalent ions around two are generally obtained independently of synthesis conditions in layered double hydroxide (LDH) structures. Conversely, the value of the *c* parameter depends on the kind of anions located in the interlayer spacing and to a lesser extent on the hydration degree. In this case, it matches LDH structures containing carbonate anions [22]. Because the X-Ray peaks associated with the wurtzite ZnO phase are overlapped with those of the Zn-Ga LDH and $ZnGa_2O_4$ phases, the diffractogram was deconvoluted to determine the corresponding lattice parameters (see Supplementary Materials). The values of the *a* and *c* lattice parameters of ZnO are in good agreement with those reported in the standard reference data (JCPDS 89-1397).

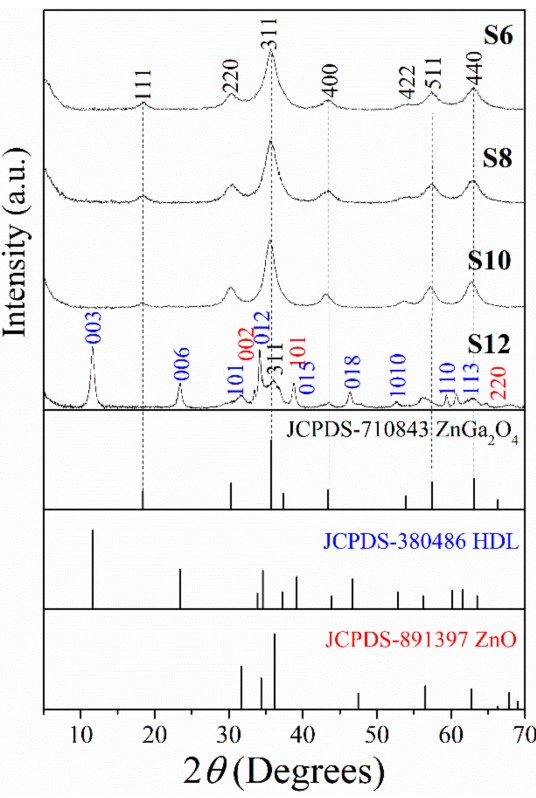

**Figure 2.** XRD patterns of the Zn-Ga samples synthesized through coprecipitation at different pH.

**Table 1.** Lattice parameters of the crystalline phases, crystallite size (*L*), and yields of the solid products obtained through coprecipitation method at different pH.

| Sample | $ZnGa_2O_4$ | | | Zn-Ga LDH | | ZnO | |
|---|---|---|---|---|---|---|---|
| | *a*, Å | *L* *, nm | Yield, % | *a*, Å | *c*, Å | *a*, Å | *c*, Å |
| S6 | 8.34 | 4.5 | 50.8 ± 8.2 | - | - | - | - |
| S8 | 8.35 | 4.3 | 36.3 ± 6.1 | - | - | - | - |
| S10 | 8.38 | 5.5 | 32.0 ± 6.78 | - | - | - | - |
| S12 | 8.46 | 5.5 | 21.3 ± 7.8 | 3.11 | 22.86 | 3.26 | 5.18 |

* Calculated from the 400 reflection.

The profile of the X-Ray reflections of the $ZnGa_2O_4$ phase suggests that the synthesis method leads to nanometric particles being obtaining. The average crystallite size calculated using the Scherrer formula for the different samples was in the range of 4.3–5.5 nm (see Table 1), which is less than half the size of those particles obtained by Safeera et al. [23] using soft synthesis conditions.

The solids yield of the synthesis presented herein is indicated in Table 1. As can be noted, the product yield decreases with the pH value. This is attributed to the redissolution of a fraction of $Ga(OH)_{3(s)}$ and $Zn(OH)_{2(s)}$ nuclei to produce $Ga(OH)^{4-}$ and $Zn(OH)_4^{2-}$ species [24].

### 3.2. Titration Curve

It is widely recognized that a titration study must be conducted to identify precipitation and coprecipitation zones of metallic cations [25–27]. The titration curve of the Zn-Ga precursor solution using KOH as precipitant agent is presented in Figure 3. From this, four sections were identified: 0–3.2, 3.2–5.0, 5.0–11.4, and 11.4–14.0. At the pH interval between 0 and 3.2, different complexes such as $Zn(H_2O)_6^{2+}$, $Ga(H_2O)_6^{3+}$ and $GaOH_2^+$ coexist in the aqueous media. Subsequently, it has been reported that mononuclear and polynuclear

species such as $ZnOH^+$, $Ga(OH)^{2+}$, and $Ga_5(OH)_9^{6+}$ can be found in the pH range between 3.2–5.0, with precipitation of amorphous $Ga(OH)_{3(s)}$ beginning simultaneously [19]. A slope change is clearly visualized at pH 5.0 and continues until around 11.4. Accordingly, it has been reported that individual gallium or zinc species such as $Ga(OH)_{3(s)}$ [19] and $GaO(OH)$ [28] form, precipitation of $Zn(OH)_2$ occurs, and $Zn(OH)_2$ partial dissolution generates $Zn(OH)^{3-}$ species [20]. Finally, in the 11.4–14.0 pH interval, the strong alkaline conditions lead to formation of mononuclear species such as $Ga(OH)^{4-}$ and $Ga(OH)_5^{2-}$ anions [21], which are the precursors of the $Ga_2(OH)_5(NO_3)$ solid phase. On the other side, $Zn(OH)_4^{2-}$ species form from $Zn(OH)^{3-}$ at this pH interval, and $Zn(OH)_2$ re-dissolution takes place [29]. It is recognized that the precipitation zones of individual metallic salts are distinct from coprecipitation of mixtures of metallic salts. For instance, Ross and Kodama [21] carried out a comparison between the precipitation of $Mg(OH)_2$ from $MgCl_2$ and $MgCl_2/AlCl_3$ solutions using NaOH as precipitant agent. They observed that $Mg(OH)_2$ precipitated at pH 9.5 from the $MgCl_2$ solution. Conversely, the mixture presented a pH plateau between 7.7 and 8.5 associated with formation of $Mg(OH)_2$. The authors attributed this buffering effect to the interaction between magnesium and aluminum species giving rise to formation of an anionic clay.

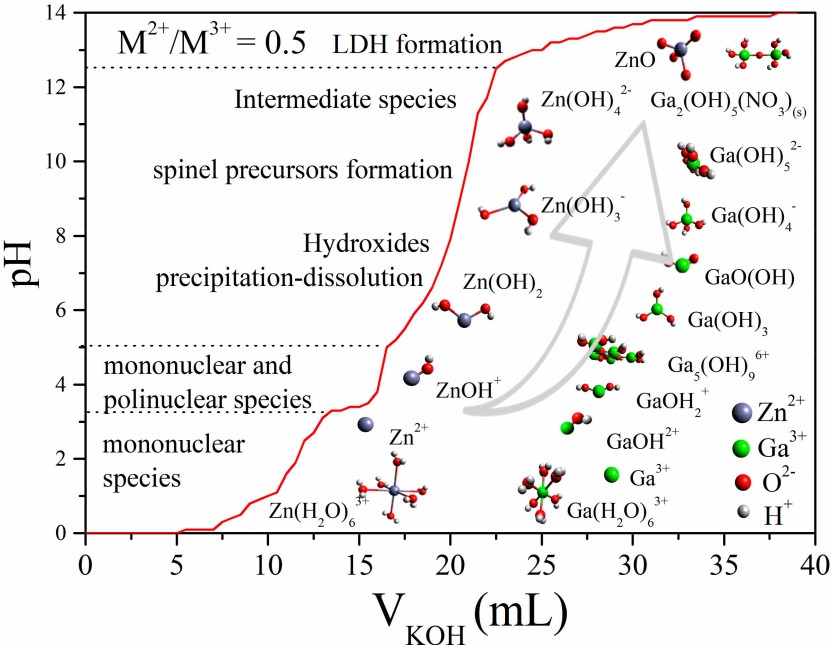

**Figure 3.** Titration curve of the Zn-Ga precursor solution (Zn:Ga 1:2 mol:mol) using a 0.5 M KOH solution as precipitant.

### 3.3. Reaction Mechanism Proposal

Titration curve analysis and identification of the distinct species is a requirement of a proposed reaction mechanism. In this sense, as the pH is kept constant in each synthesis, the reaction mechanism can be envisaged as a simultaneous coprecipitation-dissolution-crystallization process, where the precursors coprecipitate, then, depending on the species stability, re-dissolve and crystallize in the form of a more stable compound. Figure 4 presents the formation mechanism proposed for $ZnGa_2O_4$ and the anionic clay material (Zn-Ga LDH).

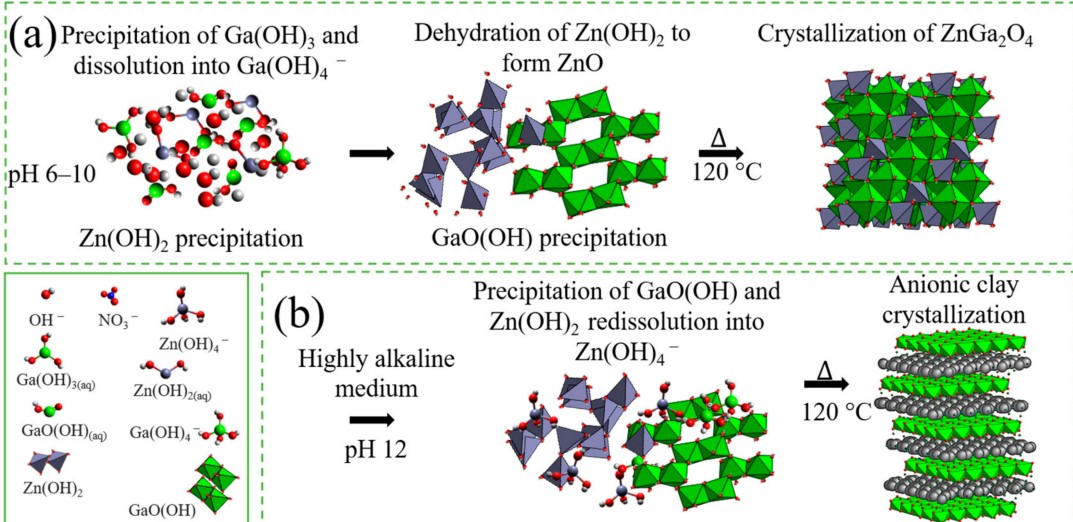

**Figure 4.** Scheme of the formation mechanism of (**a**) spinel and (**b**) anionic clay and ZnO phases by a pH-dependent coprecipitation route. The precipitation, nucleation, dehydration, and crystallization stages are shown.

For $ZnGa_2O_4$ formation, it is proposed that, in the 6–10 pH range, $Zn(OH)_2$, $Ga(OH)_3$, and $GaO(OH)$ are present in solution form along with anionic species such as $Ga(OH)_4^-$ and $Zn(OH)^{3-}$, acting as precursors of $ZnGa_2O_4$ crystals. ZnO crystallization can be attributed to either $Zn(OH)_2$ dehydration [30] or to the $Zn(OH)_3^-$, $Zn(OH)_4^{2-}$, and $[Zn_2(OH)_8(H_2O)_3]^{4-}$ decomposition–precipitation process [31]. In the case of the LDH phase, it is suggested that $Zn(OH)_4^{2-}$ anions are involved in the formation mechanism of the $Zn(OH)_2$ solid phase, then gallium atoms resulting from dissolution of $GaO(OH)$ species diffuse into $Zn(OH)_2$, leading to LDH crystallization [31].

### 3.4. SEM Analysis

Figure 5 shows SEM micrographs of the obtained materials.

In the case of the S6 sample, irregular agglomerates of particles with hemispherical shape and average size of 78 nm are observed. These particles are associated with the $ZnGa_2O_4$ phase. As the pH increases, agglomerates of similar smaller particles continue to be observed; however, their morphology is more irregular. Interestingly, several platy-like particles are identified in the S10 sample. In the micrograph of sample S12, particle agglomerates mixed with platy-like structures with lengths around 380 nm are observed. These elongated structures are associated with the characteristic habit of LDH crystals [32]. This evidence corroborates the progressive transition of spinel-like structures to LDH phase.

### 3.5. Electronic Properties

The solid samples were further studied through DRS (see Figure 6). The spectra of all samples present a strong absorption edge around 260 nm, attributed to the electronic band-to-band transition of $ZnGa_2O_4$. Excepting S12, a second absorption edge is observed around 330 nm. Because the XRD patterns of samples S6–S10 exhibit only the characteristic $ZnGa_2O_4$ reflections, this feature can be attributed to an optical absorbing trace, probably a solid solution formed by diffusion of interstitial $Zn^{2+}$ ions into the spinel lattice [33]. Because the absorption edge attributed to zinc gallate is not hidden by the optical absorption of the narrower band gap phase, the latter must be a minority phase, in agreement with the X-Ray patterns. In the case of S12, its spectrum is dominated by an optical absorption edge located around 370 nm, which is attributed to a non-negligible amount of wurtzite ZnO [34]. The residual reflectance at the shortest wavelength is associated with the Zn-Ga LDH phase, which has a bandgap energy higher than 5.5 eV [35,36].

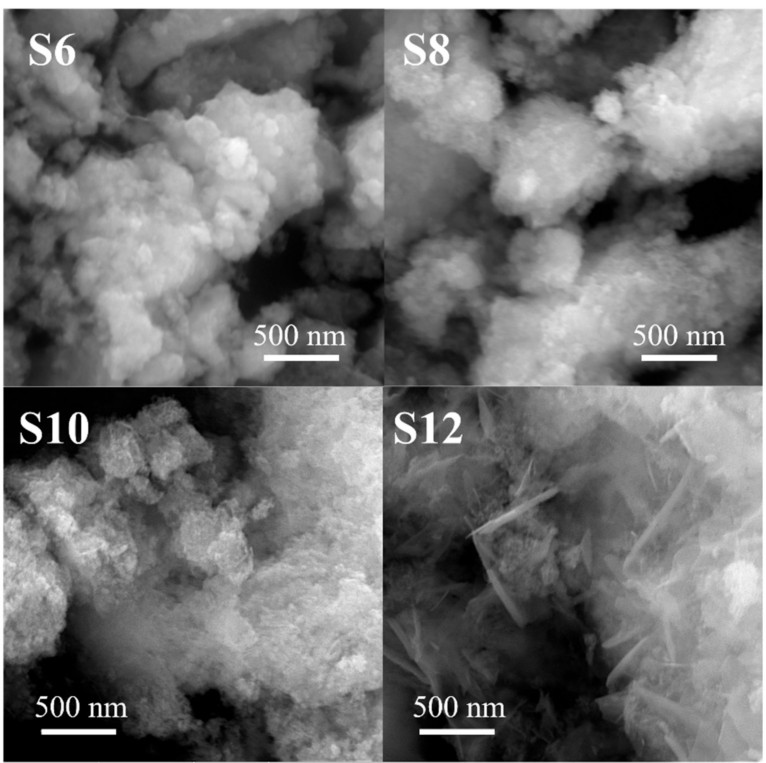

**Figure 5.** SEM micrographs of the Zn-Ga samples synthesized through coprecipitation route at different pH.

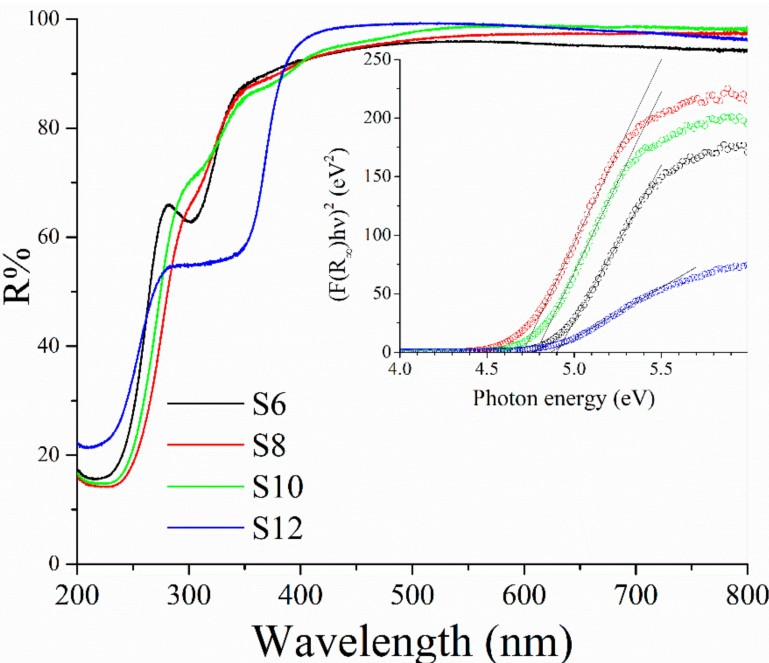

**Figure 6.** Diffuse reflectance spectra and Tauc plots (inset) of samples synthesized at different pH.

The diffuse reflectance spectra of the samples were transformed by means of the Kubelka–Munk formalism, then the bandgap energy ($E_g$) of the present phases were estimated using the Tauc method [37] (see Table 2). The $E_g$ of S6 was determined to be 4.9 eV, which agrees with the reported values (4.4–5.0 eV) of zinc gallates obtained by different synthesis methods [3,38]. It was noticed that the $E_g$ value of the zinc gallate phase decreased with the synthesis pH. This effect can be attributed to a large density of

point defects generating shallow levels into the forbidden bandgap. Similarly, the bandgap energy of the $Zn_{2x}Ga_{2-2x}O_{3+x}$ [39] phase shows a dependence on the synthesis pH. In this case, the small variation can be ascribed to the limited solubility of zinc in the gallium oxide structure along with the depletion of available zinc species that easily react to form zinc hydroxides, leading to ZnO seeding as the alkalinity of the reaction medium increases [40].

**Table 2.** Bandgap energy ($E_g$) and specific surface area (SSA) of the phases identified in the Zn-Ga samples synthesized at different pH.

| Sample | $E_g$ (eV) | | | SSA ($m^2$ $g^{-1}$) |
| :---: | :---: | :---: | :---: | :---: |
| | $ZnGa_2O_4$ | $Zn_{2x}Ga_{2-2x}O_{3+x}$ | ZnO | |
| S6 | 4.9(0) | 3.8(1) | - | 140 |
| S8 | 4.7(1) | 3.8(0) | - | 113 |
| S10 | 4.7(8) | 3.7(2) | - | 82 |
| S12 | 4.8(8) | - | 3.3(6) | * |

* Not determined; ( ) Standard deviation values.

### 3.6. Textural Properties

The textural properties of the samples are summarized in Table 2 and Figure 7. The sample prepared at pH 12 was discarded from textural analysis because it presented the lowest reaction yield and the largest amount of undesired phases (Zn-Ga LDH; ZnO). The IV(a)-type adsorption–desorption isotherms reveal that the zinc gallate samples have a mesoporous nature. The hysteresis loops indicate that capillary condensation occurs inside pores of size >4 nm [41]. The samples prepared at pH 6 and 10 displayed H3 hysteresis loops, while that synthesized at pH 8 exhibited a combination of H1 and H3 loops. In this context, the H3-type is assigned to aggregates of platy particles or materials with slit-shaped pores, whereas the H1-type is attributed to materials having narrow distribution pore size [42].

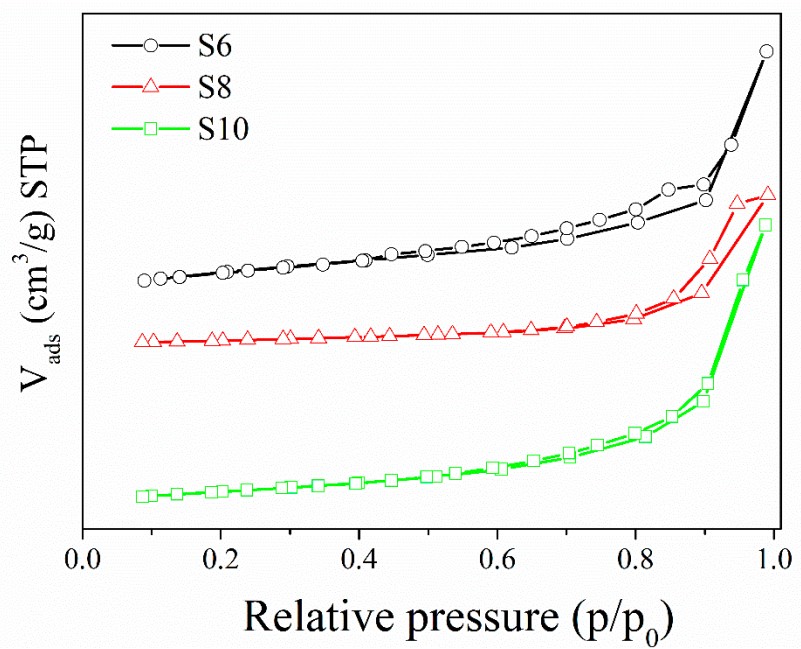

**Figure 7.** $N_2$ adsorption–desorption isotherms of S6, S8 and S10.

It is important to emphasize that the specific surface area (SSA) of materials depends on the synthesis procedure and the precursors used. For instance, Yang et al. [43] prepared mesoporous, nanoflower, and bulk $ZnGa_2O_4$, obtaining SSA values of 157, 94, and 13 $m^2$ $g^{-1}$, respectively. On the other hand, Hirano and Sakaida [44] obtained zinc gallate

samples by annealing (300–900 °C) powders synthesized through a hydrothermal method using zinc and gallium sulfates as precursors. The resulting solids achieved SSA values between 6 and 82 m$^2$ g$^{-1}$. In the same context, Zeng et al. [3] synthesized ZnGa$_2$O$_4$ spinels by three distinct methods, using ZnO, Ga$_2$O$_3$, GaOOH, and/or Zn(CH$_3$COO)$_2$·2H$_2$O as chemical precursors, attaining values between 5 and 29 m$^2$ g$^{-1}$. The SSA values of the samples prepared in this work are presented in Table 2. It can be observed that it decreases from 140 to 82 m$^2$ g$^{-1}$ as the pH increases. This behavior agrees with the crystallite sizes estimated from the XRD analysis.

### 3.7. Photocatalytic Activity

The photocatalytic activity of the samples containing zinc gallate as the majority phase were evaluated through phenol degradation under UV light irradiation. A photolysis test was carried out in order to verify phenol stability under the light source. The results revealed that phenol molecules were stable during the initial 120 min, then degraded by 28% after 360 min. Figure 8a shows the absorbance spectra and pH change of the phenol solution containing the S6 catalyst during the adsorption–desorption period. Although no significant pH variation of the suspension was measured (6.1–5.8), it was noticed that the band centered at 210 nm broadens and increased its optical absorbance after 30 min in the dark. This hyperchromic shift is associated with the influence of the OH group in the benzene ring. The presence of this chemical group increases the electron density due to the unshared electron pair of the oxygen atom, which causes conjugation between the OH group and the π-bonds of the benzene ring. As a result, the O-H polarity increases and phenol dissociates in aqueous solutions to produce phenolate and hydrogen ions [45]. It has been reported that the point zero charge (pzc) of ZnGa$_2$O$_4$ is pH 5.9. Under this condition, hydroxyl groups might bound to the ZnGa$_2$O$_4$ surface. Thus, it is possible for these groups to interact with phenol in the aqueous solution [46,47]. Figure 8b shows the evolution of phenol absorbance spectra during photocatalytic experiments in the presence of the S6 sample. In addition to the hyperchromic effect, the absorbance of the band at 270 nm decreases after 120 min of UV irradiation, and this behavior continues gradually until it disappears entirely. Additionally, a new band centered at 290 nm arises, reaching a maximum after 150 min of irradiation before its absorbance value progressively decreases to zero (240 min).

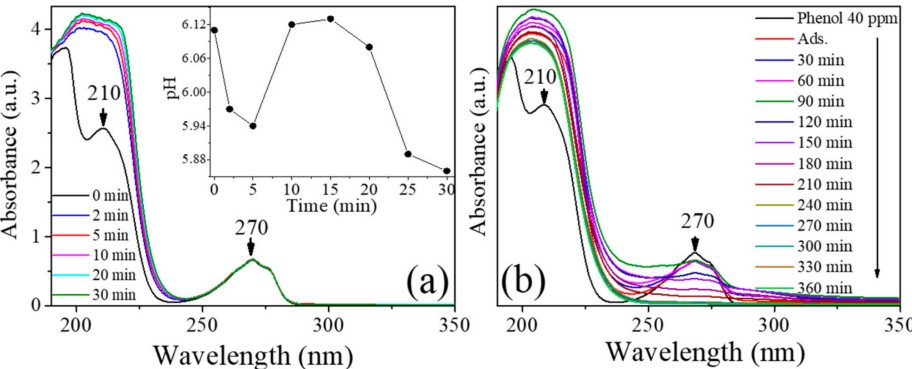

**Figure 8.** Time evolution of the absorbance spectrum of the phenol solution containing the S6 catalyst under (**a**) dark conditions (absorption–desorption) and (**b**) UV irradiation (photocatalysis). Inset: pH changes of the test solution under dark conditions.

The spectra obtained during the evaluation of the S8 catalyst (Figure 9a) presented similar behavior than that of the S6 sample. However, for S10 and TiO$_2$-Degussa P25, the band at 290 nm was detected during the adsorption–desorption experiment and after just 30 min of irradiation, respectively. This behavior suggests formation of adduct intermediates, probably generated by oxidation of phenol through addition of OH radicals to the carbon ring in different positions (ortho-, metha-, or para-). In the next step, a water molecule and a

derivative are formed through hydrogen abstraction. If a second OH radical is added, three different byproducts can be formed, namely, catechol, hydroquinone, and resorcinol [48]. Although byproduct generation was detected, their related absorbance bands are quenched progressively, suggesting that the degradation of both phenol molecules and intermediates continued, leading to formation of C-H, H-O, and C-O hydrocarbon chains. Afterwards, these compounds were further oxidized to form $CO_2$ and $H_2O$ molecules [49].

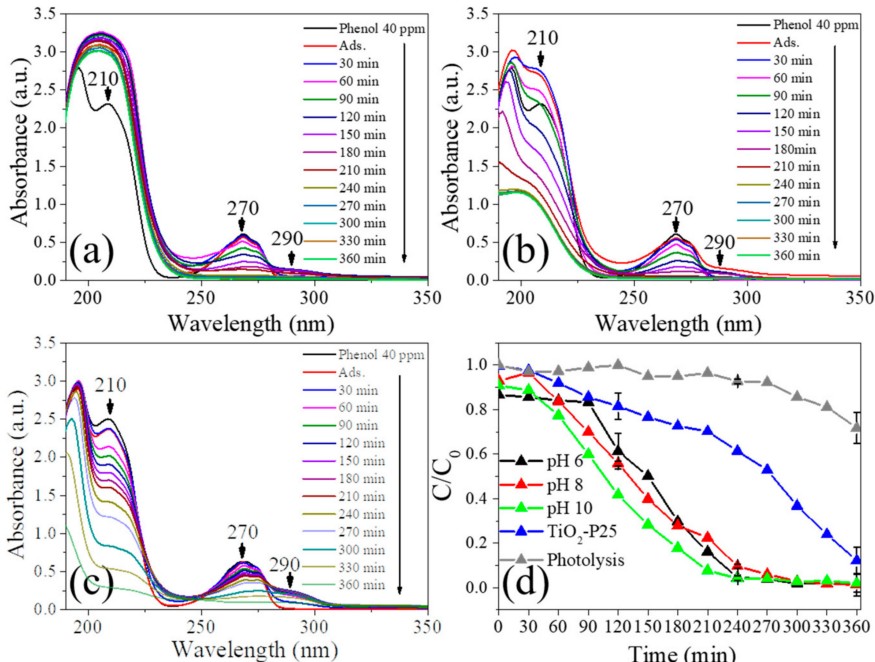

**Figure 9.** Time evolution of the absorbance spectrum of the aqueous phenol solution using (**a**) S8, (**b**) S10, and (**c**) TiO$_2$-Degussa P25 catalysts. (**d**) Phenol degradation curves using the samples synthesized at different pH as photocatalysts.

Figure 9d displays the concentration changes of phenol for the selected samples. In general, S6, S8, and S10 present similar photocatalytic activity. For comparison purposes, the phenol degradation percentages were determined at distinct reaction times (see Table 3). After 120 min of UV irradiation sample S10 degraded 58.6% of the initial phenol, whereas for samples S8 and S6, 44.5% and 38.6% was degraded, respectively. Analogous results were observed among the spinel-containing samples at longer reaction times, achieving phenol degradation between 90–95% and ~98% after 240 and 360 min, respectively. In this regard, the TiO$_2$-Degussa P25 standard degraded significantly lower amounts of phenol regardless of the elapsed time, demonstrating that the samples prepared by the method presented herein have better catalytic performance than commercial TiO$_2$ under the same test conditions. The apparent rate constant for phenol degradation was calculated according to the integrated pseudo-first order kinetics equation:

$$\ln\left(\frac{C_0}{C}\right) = kt \tag{2}$$

where $C_0$ is the initial phenol concentration, $C$ corresponds to its concentration at time $t$, and $k$ is the apparent rate constant [50]. The values of the latter parameter for samples S6, S8, S10, and TiO$_2$-Degussa P25 were determined to be $12.3 \times 10^{-3}$, $11.9 \times 10^{-3}$, $11.6 \times 10^{-3}$, and $4.2 \times 10^{-3}$ min$^{-1}$, respectively (see Figure 10a). Because the rate constant $k$ is related to the molar concentration of the reactants, these results suggest rapid degradation of phenol in the presence of S6. Its rate constant is 2.9 times larger than that of TiO$_2$-Degussa P25.

**Table 3.** Phenol degradation percentage at different time intervals of the Zn-Ga samples and commercial TiO$_2$-Degussa P25.

| Time (min) | Phenol Degradation (%) | | | | |
|---|---|---|---|---|---|
| | Photolysis | S6 | S8 | S10 | TiO$_2$-Degussa P25 |
| 120 | 0 | 38.6(8) | 44.5(1) | 58.6(0) | 18.1(6) |
| 240 | 7.2(1) | 95.4(1) | 90.4(1) | 95.8(1) | 38.4(0) |
| 360 | 28.1(7) | 98.1(4) | 98.6(5) | 98.0(4) | 87.7(6) |

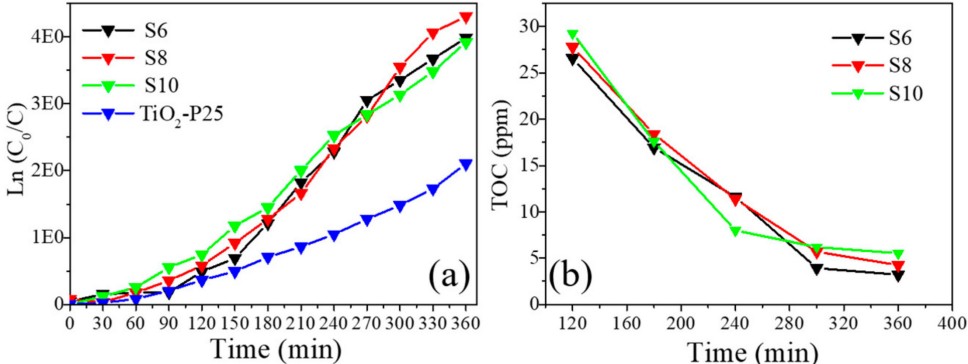

**Figure 10.** (**a**) Kinetic degradation of phenol solutions and (**b**) TOC removal curves using the samples synthesized at different pH as photocatalysts.

### 3.8. TOC Results

Although phenol degradation was achieved using the Zn-Ga samples as photocatalysts, it was necessary verify whether the degradation products were being mineralized; therefore, TOC measurements were conducted (see Figure 10b). The gradual decreases in TOC concentration after UV irradiation were $3.2 \times 10^{-3}$, $4.2 \times 10^{-3}$, and $5.5 \times 10^{-3}$ g L$^{-1}$ for samples S6, S8, and S10, respectively. The decay in TOC values confirmed phenol mineralization. Considering the TOC and photodegradation results, it was identified that the optimal sample was that synthesized at pH 6, which presented the highest product yield and largest specific surface area as well. Additionally, its preparation was the least expensive, as it required the lowest amount of precipitant agent.

### 3.9. Photocatalytic Mechanism

The suggested photocatalytic mechanism is depicted in Figure 11. This process begins when e$^-$/h$^+$ pairs are photogenerated in the ZnGa$_2$O$_4$ phase by optical absorption of photons with energy higher than or equal to its band gap (4.7–4.9 eV). Because the valence band (VB) of ZnGa$_2$O$_4$ is more positive (3.13 eV vs. NHE (normal hydrogen electrode)) than E(OH$^\bullet$/OH$^-$) (2.38 eV vs. NHE) [51] and its conduction band (CB) is more negative ($-1.27$ eV vs. NHE) than E(O$_2$/O$_2$$^{-\bullet}$) ($-0.33$ eV vs. NHE) [52], both OH$^\bullet$ and O$_2$$^{-\bullet}$ radicals can be generated. In presence of the Zn$_{2x}$Ga$_{2-2x}$O$_{3+x}$ phase, which has a narrower bandgap (3.7–3.8 eV) than zinc gallate and a more positive VB than E(OH$^\bullet$/OH$^-$), the heterojunction creates an effective pathway for charge separation, enhancing the formation of OH$^\bullet$ radicals, which in turn boosts the photocatalytic activity.

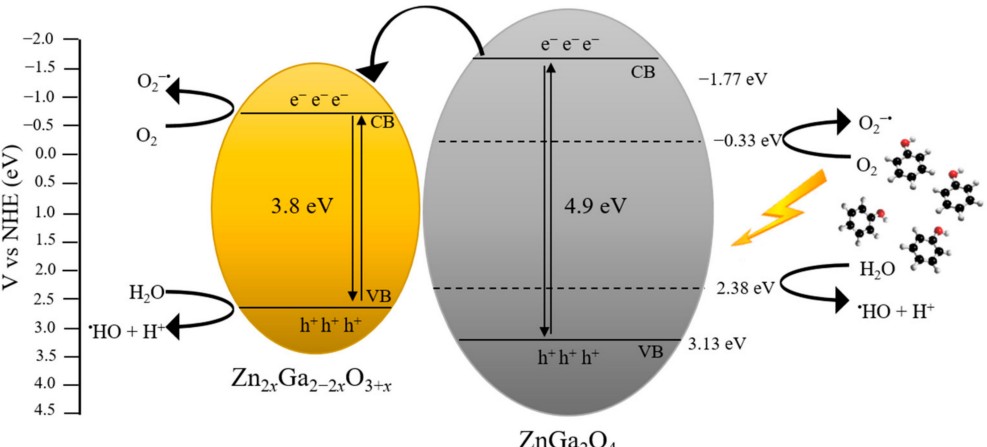

**Figure 11.** Schematic illustration of the proposed photocatalytic mechanism of $Zn_{2x}Ga_{2-2x}O_{3+x}/ZnGa_2O_4$.

## 4. Conclusions

The coprecipitation method proposed herein is presented as a fast and simple alternative to solid-state reactions for obtaining crystalline $ZnGa_2O_4$ powders. Using this chemical approach, either $ZnGa_2O_4$ (pH 6–10) or $ZnGa_2O_4$/Zn-Ga LDH/ZnO mixtures (pH 12) can be synthesized by adjusting the pH of the reaction medium. Moreover, this synthesis route requires mild reaction conditions and short aging times (25 °C; 30 min). It has been demonstrated that the pH has an important effect on the formation mechanism and the reaction product yield, determining the generation of specific mononuclear and polynuclear species that favor formation of definite phases such as spinel $ZnGa_2O_4$ or Zn-Ga layered double hydroxide (an anionic clay). In this sense, the largest product yield was achieved at pH 6 (50.8 ± 8.2%), with yields decreasing under strong alkaline conditions (pH 12; 21.3 ± 7.8%). This effect was attributed to the redissolution of a fraction of $Ga(OH)_3$ and $Zn(OH)_2$ nuclei to produce $Ga(OH)_4^-$ and $Zn(OH)_4^{2-}$ species. It was verified that the morphology, textural and electronic and photocatalytic properties of the obtained materials can be tuned by controlling the synthesis pH. It was further shown that the coprecipitated $ZnGa_2O_4$ powders can be used successfully as photocatalysts, with the sample obtained at pH 6 able to degrade 98.1% of phenol dissolved in aqueous media and reaching up to 93.0% of mineralization after 6 h. Finally, it is proposed that the reported coprecipitation approach can be used to obtain other spinel-like compounds while requiring only short reaction times and low energy investment.

**Supplementary Materials:** The following supporting information can be downloaded at: https://www.mdpi.com/article/10.3390/cryst13060952/s1, Figure S1: Detail of the 400 X-ray peak attributed to ZnGa2O4 spinel synthesized at different pH, Figure S2: Deconvolution of the diffractogram of the S12 sample.

**Author Contributions:** Conceptualization: M.S.-C., A.E.-M. and D.T.-F.; Methodology: M.S.-C., A.E.-M. and D.T.-F.; Validation: M.S.-C., A.E.-M., D.T.-F., M.d.L.R.-P., E.L.-S., F.T. and A.P.-C.; Formal analysis: M.S.-C., A.E.-M. and M.d.L.R.-P.; Investigation: M.S.-C., A.E.-M. and D.T.-F.; Resources: M.S.-C.; Data curation: M.S.-C., A.E.-M. and M.d.L.R.-P.; Writing—original draft: M.S.-C., A.E.-M., M.d.L.R.-P. and D.T.-F.; Writing—review and editing: M.S.-C., A.E.-M., D.T.-F., M.d.L.R.-P., E.L.-S., F.T. and A.P.-C.; Visualization: M.S.-C., D.T.-F., A.E.-M. and M.d.L.R.-P.; Supervision: M.S.-C. and A.E.-M.; Project administration: M.S.-C.; Funding acquisition: M.S.-C. and A.E.-M. All authors have read and agreed to the published version of the manuscript.

**Funding:** The APC was funded by Vicerrectoría de Investigación y Estudios de Posgrado-Benemérita Universidad Autónoma de Puebla. D.T.-F acknowledges the financial support provided by the Mexican Council of Science and Technology (CONACYT) through scholarship #732563.

**Data Availability Statement:** The data presented in this study are available on request from the corresponding authors.

**Acknowledgments:** The authors thank Efrain Rubio for his assistance in materials characterization.

**Conflicts of Interest:** The authors declare no conflict of interest.

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
