# Peer review of "Aqueous Chemical Synthesis of Nanosized ZnGa2O4 Using Mild Reaction Conditions: Effect of pH on the Structural, Morphological, Textural, Electronic, and Photocatalytic Properties"

_crystals, doi:10.3390/cryst13060952_

Round 1

Reviewer 1 Report

This work provides the synthesis of ZnGa2O4 powders by aqueous soft coprecipitation of zinc and gallium nitrates under acid or basic conditions applying short ageing times. The effect of the synthesis pH on the microstructure, morphology, electronic and textural properties of the obtained materials were investigated together with their photocatalysts using phenol as probe molecule. According to this research line, this manuscript well provides reliable data. These data are valuable of being published in well-authorized scientific journals. Therefore, I may recommend publication of this work in Crystals. However, this manuscript can be improved by some revisions on general points. Please see below.

1) The current title uses rather common and technical words. In such case, inclusion of new concept words would give more innovative impression. I may suggest use of an emerging conceptual term, nanoarchitectonics, in the title (as post-nanotechnology concept, see https://pubs.rsc.org/en/content/articlelanding/2021/nh/d0nh00680g). For example, the title like ... Aqueous chemical nanoarchitectonics of spinel ZnGa2O4 using mild re-2 action conditions: effect of pH on the structural, morphological, 3 textural, electronic and photocatalytic properties ... may sound more innovative.

2) Please add the initial figure to explain the used materials and their synthetic methods.

3) Word sizes in figures are much different between figures. Even thou perfect adjustment would not be possible, please try to unify word sizes as good as possible.

4) Descriptions of conclusion are rather short and insufficient. More detailed descriptions including future perspectives had better be added.

Author Response

Q1. The current title uses rather common and technical words. In such case, inclusion of new concept words would give more innovative impression. I may suggest use of an emerging conceptual term, nanoarchitectonics, in the title (as post-nanotechnology concept, see https://pubs.rsc.org/en/content/articlelanding/2021/nh/d0nh00680g). For example, the title like ... Aqueous chemical nanoarchitectonics of spinel ZnGa2O4 using mild reaction conditions: effect of pH on the structural, morphological, textural, electronic and photocatalytic properties may sound more innovative.

A1. Although including the term nanoarchitectonics could make the article’s title more attractive, the reported synthesis method does not intend to obtain some particular configuration/morphology, but the main goal is to develop a facile chemical route to produce nanosized materials. The term nanosized was included in the tile to give more innovative impression instead (p. 1).

Q2. Please add the initial figure to explain the used materials and their synthetic methods.

A2. A flowchart of the synthesis method was incorporated in the revised version of the manuscript (see Figure 1, p. 3).

Q3. Word sizes in figures are much different between figures. Even thou perfect adjustment would not be possible, please try to unify word sizes as good as possible.

A3. The font size was adjusted in the figures to gain visual uniformity.

Q4. Descriptions of conclusion are rather short and insufficient. More detailed descriptions including future perspectives had better be added.

Q4. The conclusions section was edited including detailed descriptions, quantitative data and future perspectives (p. 14).

Reviewer 2 Report

The presented manuscript seems to be interesting for readers of the crystals journal, it is written in a good manner and suits the requirements of the journal. It can be accepted for publication after the minor corrections listed below.

- The "Abstract" section should contain the main achievements of the research, not a general discussion. Re-organization of the abstract is needed.

- The novelty of the work at the end of the manuscript “introduction” is not sufficient and should be explained more.

- Abbreviations/ acronyms, should all be defined at their first occurrence in the manuscript, 

- Specifications of devices and standards used in different tests should be given in the materials and methods section.

- The different parts of Figure 4 should be explained in the caption

- The optimal sample based on properties and production costs should be introduced.

- The flowchart of the research method should be given. Also, sample coding and specifications should be provided in the table.

- It is recommended to perform additional characterization (such as FTIR, Raman spectroscopy, DTA/TG) to check the effect of input parameters on the manufactured product.

- In the "Conclusion" section, the authors should present more quantitative data as the main results of the research study rather than just some qualitative data.

-The optimum condition in the fabrication operation needs to be determined. The authors need to pay attention in the revision stage.

- Literature review is not sufficient and authors must review and cite more papers in the field of “Synthesis of materials by sol-gel method” especially newly published ones. Doing this, reviewing the following refs could be helpful: Journal of Sol-Gel Science and Technology 103, no. 1 (2022): 87-96., Journal of Materials Research and Technology 22 (2023): 2462-2472.

Moderate editing of the English language required

Reviewer 3 Report

This article is comprehensive, logically organized, and contains valuable information on the effect of pH on the structural, morphological, textural, electronic, and photocatalytic properties of the aqueous chemical synthesis of spinel ZnGa2O4 using mild reaction conditions. The authors did excellent research on investigating the pH effect on the structural, textural, morphological and electronic properties of ZnGa2O4 spinel-like compounds obtained by the co-precipitation method using mild reaction conditions. The authors demonstrated that pH has an important effect on resultant phases since it is related to the generation of specific mononuclear and polynuclear species that favor not only definite crystalline phases but reaction product yield. The authors presented the phenol degradation percentage at different time intervals of the Zn-Ga samples and commercial TiO2-Degussa P25 in Table 3. This manuscript does not contain much of an error analysis on the phenol degradation performance which is highly required for readability purposes. It is suggested the authors should place the standard deviations of the phenol degradation performance data as presented in Table 3, for the reliability and readability of the present research. The submitted manuscript has significant scientific insights and the conclusions are soundly supported by the experimental data. However, the current submission requires minor revisions before being considered for publication in the esteemed Crystals in its current condition.

The pH effect on the structural, textural, morphological, and electronic properties of ZnGa2O4 spinel-like compounds obtained by the co-precipitation method using mild reaction conditions was studied. The pH ranges where co-precipitation reactions occur and the chemical species associated with the reaction mechanism were identified. The X-ray diffraction data showed that the samples synthesized at pH values between 6-10 consist of ZnGa2O4-ZnO mixtures being spinel the majority phase. Conversely, the material prepared at pH 12 showed mainly hydrotalcite-like characteristic reflections. By UV-Vis diffuse reflectance spectroscopy characteristic bandgap values were calculated. A coprecipitation-dissolution-crystallization mechanism was proposed from all the evidence obtained. Phenol was employed as a probe molecule to evaluate the photocatalytic performance of samples. The photocatalytic evaluation indicated that the sample prepared at pH6, which presented the highest yield (51%) and high specific surface area (140 m2 g-1), was able to degrade ~98% phenol with a 93% mineralization.

Author Response

Q1. This manuscript does not contain much of an error analysis on the phenol degradation performance which is highly required for readability purposes. It is suggested the authors should place the standard deviations of the phenol degradation performance data as presented in Table 3, for the reliability and readability of the present research.

A1. We appreciate the reviewer’s suggestion. Cause the complexity of the photocatalytic experiments, we opted to perform kinetic studies with large density points. Nonetheless, only for those samples indicated in Figure 9 (p. 12), additional spectrophotometric measurements were taken at the intervals 120, 240 and 360 min. It was observed that the standard deviation was negligible, indicating that the acquired data is representative, and the methodology is reliable. Considering the latter, the rest of treatments were conducted once.

Reviewer 4 Report

The presented manuscript includes the study of aqueous chemical synthesis of spinel ZnGa2O4 using mild reaction conditions: effect of pH on the structural, morphological, textural, electronic and photocatalytic properties.

The results of the work are presented on a good level, the text is well structured, and the figures are clear and well treated, but some questions and weak points regarding paper structure should be mentioned. The paper is of interest and fits the Crystals journal.

1.   Please name first paragraph in section 2. “Materials and methods” as 2.1 “Materials and reagents”.  There mention all reagents used in your study (as well as phenol), their purity and supplier.

2. Isotherms shown in Fig. 6 are incorrectly classified as Type IV. This type is divided into two: a and b. Please have a look at the IUPAC recommendations for describing isotherms (Pure Appl. Chem. 2015, 87, 1051 DOI:10.1515/pac-2014-1117) and revise the classification.

3. Please provide conclusions with exact numbers.

4. please, change ppm concentrations to SI units: mass (e.g. g/L) or molar (e.g. mol/L)

5. Most of references are too old. Please, up to date the relevant references for the 2022 and 2023 years to prove the novelty and actuality.

Author Response

REVIEWER #4

Q1. Please name first paragraph in section 2. “Materials and methods” as 2.1 “Materials and reagents”. There mention all reagents used in your study (as well as phenol), their purity and supplier.

A1. The subsection title was included, and the specifications of reagents were indicated in the revised version (p. 2).

Q2. Isotherms shown in Fig. 6 are incorrectly classified as Type IV. This type is divided into two: a and b. Please have a look at the IUPAC recommendations for describing isotherms (Pure Appl. Chem. 2015, 87, 1051 DOI:10.1515/pac-2014-1117) and revise the classification.

A2. We thank the reviewer’s observation. The classification of the isotherms was corrected following the IUPAC recommendations (p. 10).

Q3. Please provide conclusions with exact numbers.

A3. The conclusions section was edited including detailed descriptions, quantitative data and future perspectives (p. 14).

Q4. Please change ppm concentrations to SI units: mass (e.g. g/L) or molar (e.g. mol/L)

A4. The units were modified (pp. 3, 13).

Q5. Most of references are too old. Please, up to date the relevant references for the 2022 and 2023 years to prove the novelty and actuality.

A5. An extended literature review was carried out to update the related information. In the revised version, very recent works dealing with the importance of pH on controlling the properties of materials, like that published by Prof. A. Najafy’s group (J. Mater. Res. Technol. 22 (2023) 2462), were cited.

Round 2

Reviewer 2 Report

The revised manuscript is acceptable

Acceptable

Author Response

Thank you.

Reviewer 4 Report

all comments were addressed 

Author Response

Thank you.